# Delineating maternal influence in regulation of variance in major economic traits of White Leghorns: Bayesian insights

**Aneet Kour\***, **R. N. Chatterjee, K. S. Rajaravindra, L. Leslie Leo Prince**, **Santosh Haunshi, M. Niranjan, B. L. N. Reddy, U. Rajkumar** *

Poultry Genetics and Breeding Division, ICAR-Directorate of Poultry Research, Hyderabad, Telangana, India

* ullengala@yahoo.com (UR); aneet.kour@icar.gov.in (AK)

## Abstract

Proper variance partitioning and estimation of genetic parameters at appropriate time interval is crucial for understanding the dynamics of trait variance and genetic correlations and for deciding the future breeding strategy of the population. This study was conducted on the same premise to estimate genetic parameters of major economic traits in a White Leghorn strain IWH using Bayesian approach and to identify the role of maternal effects in the regulation of trait variance. Three different models incorporating the direct additive effect (Model 1), direct additive and maternal genetic effect (Model 2) and direct additive, maternal genetic and maternal permanent environmental effects (Model 3) were tried to estimate the genetic parameters for body weight traits (birth weight, body weight at 16, 20, 40 and 52 weeks), Age at sexual maturity (ASM), egg production traits (egg production up to 24, 28, 40, 52, 64 and 72 weeks) and egg weight traits (egg weight at 28, 40 and 52 weeks). Model 2 and Model 3 with maternal effects were found to be the best having the highest accuracy for almost all the traits. The direct additive genetic heritability was moderate for ASM, moderate to high for body weight traits and egg weight traits and low to moderate for egg production traits. Though the maternal heritability ($h^2_{mat}$) and permanent environmental effect ($c^2_{mpe}$) was low (<0.1) for most of the traits, they formed an important component of trait variance. Traits like egg weight at 28 weeks (0.14±0.06) and egg production at 72 weeks (0.13±0.07) reported comparatively higher values for $c^2_{mpe}$ and $h^2_{mat}$ respectively. Additive genetic correlation was high and positive between body weight traits, between egg weight traits, between consecutive egg production traits and between body weight and egg weight traits. However, a negative genetic correlation existed between egg production and egg weight traits, egg production and body weight traits, ASM and early egg production traits. Overall, a moderate positive genetic correlation was estimated between ASM and body weight traits and ASM and egg weight traits. Based on our findings, we can deduce that maternal effects constitute an important source of variation for all the major economic traits in White Leghorn and should be necessarily considered in genetic evaluation programs.

**Data Availability Statement:** Raw data file comprising of egg production and other economic traits data and pedigree file of White Leghorn

population used in this study cannot be shared publicly. This is due to the reason that there are legal restrictions for sharing the data as the data pertains to the breeding experiment and not available for open access sharing. Furthermore, since it contains information regarding the selection of population and other important information, it is not supposed to be shared in raw form. These restrictions are imposed by the Research Advisory Council of the institute (ICAR-Directorate of Poultry Research, Hyderabad) where the data has been generated and used for selection and genetic improvement. Such data is only available upon request (for validation purposes, if any) from the corresponding authors (ullengala@yahoo.com; ANEET.KOUR@icar.gov. in). The institutional contact for data request can also be made to the Director, ICAR-DPR, Hyderabad. The contact detail is: Director. dpr@icar.gov.in.

**Funding:** The author(s) received no specific funding for this work.

**Competing interests:** The authors do not have any competing interests

## Introduction

Intense selection for higher genetic gain is the hallmark of poultry breeding programs throughout the world [1, 2]. However, it causes negative linkage disequilibrium between the loci influencing the trait and leads to a reduction in the additive genetic variance (Bulmer effect) for the trait under selection [3, 4]. This further gets aggravated in case of small and closed populations (like inbred lines) where strong directional selection in a trait over the generations can reduce the additive component of variance substantially and effectively leaves little for selection to act upon [5]. Reduction in additive genetic variance will invariably affect the trait heritability and estimated breeding values (EBVs) of the individuals for the trait [6]. Furthermore, in case of multi-trait selection, long-term selection can even change the genetic correlations between the traits due to fixation of positively pleiotropic genes in the population and dominance of negative pleiotropy [4]. Therefore, computation and monitoring of genetic parameters of a population from time to time is crucial to study the dynamics of genetic variance and to obtain appropriate selection response. It also helps in optimization of breeding programs by indicating whether there is a need for introduction of genetic variation from outside [7].

Accuracy of the estimated genetic parameters is contingent on the proper variance partitioning for the trait under consideration [8, 9]. The true picture of the additive genetic worth of a population for a trait will come to light once the variance attributable to every possible source of variation *viz.* maternal genetic or indirect genetic effect and maternal permanent environment is properly accounted for in the model [10]. Ignoring these effects in variance estimation can produce seriously biased estimates and can adversely affect the genetic improvement program [11–13]. Moreover, in poultry, many of the traits like age at first egg, birth weight and early body weights and egg weights etc. are maternally influenced traits i.e. there is a low to moderate maternal heritability component involved in the total trait variance [14–17]. Sometimes, the maternal variance component for the traits even exceeds the direct additive component [18]. Therefore, proper partitioning of a trait variance can minimize uncertainty or error variance and prevents the overestimation of additive genetic component [19, 20].

Another important factor influencing the accuracy of genetic parameters is the sample size. Larger the sample size, more precise will be the genetic estimates [21, 22]. Whenever sample size is less, the traditional likelihood-based analysis like REML (Restricted Maximum Likelihood) can give rise to biased genetic estimates [23]. In such cases, Bayesian sampling can more effectively model data with smaller effective population sizes given that proper priors are specified [24]. In Bayesian approach, prior information is effectively integrated with the data likelihood to create a posterior distribution of samples from which inferences are drawn [25]. Thus, as compared to the REML approach, it improves the accuracy of genetic estimates obtained from small populations. These days, Bayesian methods are being increasingly used to derive accurate genetic estimates in native chicken breeds as well as in commercial lines [26–28].

Since proper variance partitioning and sample size are two important considerations for improving the accuracy of genetic estimates, an effective strategy would be to estimate additive genetic and maternal variance components for the traits using Bayesian estimation. Further, we hypothesize that maternal genetic and permanent environmental effects play an important role in regulating trait variance in chickens. Therefore, this study was designed to appropriately partition the variance components and to estimate the genetic parameters of major economic traits in a population of Indian White Leghorn by trying different BLUP animal models using Bayesian approach.

## Materials and methods

### Study details

The present investigation was performed in the IWH line of White Leghorn layer breed maintained at ICAR-Directorate of Poultry Research, Hyderabad, Telangana, India. This line has been developed by selecting parents for higher egg production up to 64 weeks of age using Osborne Index. Pedigreed mating in full sib design with 50 sires and 250 dams (mating ratio of 1:5) is being followed for regeneration. A total of 1434 chicks belonging to three (3) generations (2020–23) and five (5) hatches were considered for this study. Chicks were wing banded at the time of pedigree hatch and reared in a deep litter brooder house till they attained 16 weeks of age. At six weeks, sexing was done to separate males and females. Layer starter diet (0–6 weeks) based on maize and soybean meal with 18% crude protein and 2800 kcal/kg ME was fed *ad libitum* to the chicks up to 16 weeks of age. Thereafter, layer grower ration was fed from 6–16 weeks of age which comprised of 18% crude protein and 2700 kcal/kg ME. After 16 weeks, adult males and females were transferred to individual cages and were fed layer breeder ration with 16.5% crude protein, 2650 kcal/kg ME and 3.5% calcium. In the layer cage house, daily allowance of eight hours of day light and seven hours of artificial light was provided. Since the farm is situated in the Deccan plateau region of India at an elevation of 500 metres above mean sea level (geographical coordinates 17˚23' N and 78˚ 28' E), the place experiences hot and humid tropical climate. Therefore, in order to avoid heat stress during summers, ambient temperature was maintained in the layer house through sprinklers installed on roof top. Also, a standard vaccination schedule was followed in the farm and birds were vaccinated against all the major diseases: Marek's disease (Day 1), New Castle Disease, Lasota (Day 6 and Day 28), Infectious Bursal Disease (Day 14 and Day 24), Fowl Pox (Day 42), New Castle Disease, $R_2B$ (Day 65), Infectious Coryza (Day 84) and Triple killed vaccine (Infectious Bursal Disease, Infectious Bronchitis and New Castle Disease inactivated) (112th day and 45th week). The recorded mortality in the flock was less than 5%.

This study was approved by Institutional Animal Ethics Committee of ICAR-DPR vide IAEC/DPR/20/7.

### Traits recorded

Data spanning three generations from 2020–23 was recorded for different production and growth traits up to 72 weeks of age in the female IWH population. First of all, body weight of chicks at the time of hatch (BW0) was recorded. Thereafter, when female birds were housed in individual cages, body weight at different intervals *viz*. 16 (BW16), 20 (BW20), 40 (BW40) and 52 (BW52) weeks was successively recorded. Body weights were measured to 0.1 g accuracy using a digital weighing balance. Reproduction trait like age at sexual maturity (ASM) was also recorded for all the birds.

Production data comprised of egg production or egg numbers up to 24 (EP24), 28 (EP28), 40 (EP40), 52 (EP52), 64 (EP64) and 72 (EP72) weeks and egg weights at 28 (EW28), 40 (EW40) and 52 (EW52) weeks of age. Egg production of individual birds was recorded daily and egg production up to a certain age was arrived at by summing the daily production up to that age. Egg production data after 40 weeks was not available for the third generation. Therefore, EP52, EP64 and EP72 data used to derive the genetic parameter estimates belonged to two generations only.

Further, egg weight at a particular age was estimated by taking the average of egg weight measurements of five consecutive days at that age. For instance, at 28 weeks of age, egg weight was recorded for five successive days for each bird and the average of those five measurements was finally considered as EW28.

**Table 1. Pedigree and data summary of the study population.**

| Trait | No. of birds with records | No. of sires | No. of dams | Mean±S.E. | S.D. | C.V. (%) |
|---|---|---|---|---|---|---|
| ASM | 1319 | 117 | 347 | 137.56±0.27 | 8.94 | 6.50 |
| BW0 | 936 | 88 | 220 | 35.05±0.11 | 3.35 | 9.56 |
| BW16 | 1064 | 118 | 340 | 973.31±3.83 | 124.61 | 12.80 |
| BW20 | 1145 | 72 | 223 | 1217.23±4.15 | 140.32 | 11.53 |
| BW40 | 995 | 115 | 334 | 1480.83±6.15 | 194.06 | 13.10 |
| BW52 | 709 | 71 | 215 | 1516.04±7.13 | 189.92 | 12.53 |
| EW28 | 1041 | 116 | 336 | 47.77±0.09 | 3.06 | 6.41 |
| EW40 | 937 | 115 | 329 | 51.34±0.11 | 3.34 | 6.50 |
| EW52 | 662 | 71 | 213 | 54.17±0.16 | 4.16 | 7.68 |
| EP24 | 1051 | 106 | 318 | 26.13±0.23 | 7.19 | 27.54 |
| EP28 | 1046 | 110 | 324 | 49.11±0.33 | 10.54 | 21.46 |
| EP40 | 1021 | 110 | 320 | 117.56±0.58 | 18.64 | 15.86 |
| EP52 | 669 | 72 | 193 | 188.59±0.77 | 19.91 | 10.56 |
| EP64 | 603 | 70 | 189 | 251.70±1.04 | 25.57 | 10.16 |
| EP72 | 546 | 69 | 180 | 283.49±1.26 | 29.43 | 10.38 |

Recorded data was thoroughly screened for the removal of outliers and observations falling within Mean ± 3S.D. of normal distribution were retained for further analysis. The data structure of the study population for the traits has been presented in Table 1.

## Statistical analysis

Generation (N = 3) and hatch (N = 5) were identified as the non-genetic (fixed) factors influencing the traits. Three BLUP animal models incorporating one or more of the variance components (direct additive, maternal genetic and maternal permanent environmental effects) along with fixed effects and residual were tried to identify the best fitted model for a trait.

The linear mixed models used are as follows:

$$\mathbf{y} = \mathbf{Xb} + \mathbf{Zu} + \mathbf{e} \tag{Model1}$$

where y = a vector of observations (or records) for the trait, b = vector of fixed effects for the trait, u = vector of random additive genetic effect for the trait, e = vector of random residual effects, X = incidence matrix relating observations to fixed effects, Z = incidence matrix relating records to random animal effects.

Here, $\mathrm{var}(u) = A\sigma^2_u = G$
$\mathrm{var}(e) = I\sigma^2_e = R$
where A = Numerator relationship matrix, I = Identity matrix
In this case, variance for a trait is given by: var(y) = var(Zu + e)
$$\mathrm{var}(y) = ZGZ' + R$$
$$\mathrm{var}(y) = ZGZ' + I\sigma^2_e$$

$$\mathbf{y} = \mathbf{Xb} + \mathbf{Zu} + \mathbf{Wd} + \mathbf{e} \tag{Model2}$$

where d = vector of random maternal (indirect) genetic effect, W = incidence matrix relating records to maternal genetic effects; rest of the notations same as mentioned above.

Since there is covariance between additive genetic effect and maternal genetic effect, the combined variance for both the effects is:

$$\mathrm{var}\begin{pmatrix} u \\ d \end{pmatrix} = [gA]$$

where, $g = \begin{pmatrix} g_{11} & g_{12} \\ g_{21} & g_{22} \end{pmatrix}$

$g_{11}$ = additive genetic variance for direct effects, $g_{22}$ = additive genetic variance for maternal effects, $g_{12}$ = additive genetic covariance between direct and maternal effects

In this case, variance for a trait is given by:

$$var(y) = [Z\ W] \begin{pmatrix} g_{11}A & g_{12}A \\ g_{21}A & g_{22}A \end{pmatrix} \begin{pmatrix} Z \\ W \end{pmatrix} + I\sigma^2_e$$

$$\mathbf{y = Xb + Zu + Wd + Mpe + e} \tag{Model3}$$

where pe = vector of random maternal permanent environmental effect, M = incidence matrix relating records to maternal permanent environmental effect, rest of the notations same as mentioned above

Here, $var(pe) = I\sigma^2_{pe}$

In this case, variance for a trait is given by:

$$var(y) = [Z\ W] \begin{pmatrix} g_{11}A & g_{12}A \\ g_{21}A & g_{22}A \end{pmatrix} \begin{pmatrix} Z \\ W \end{pmatrix} + MI\sigma^2_{pe}M' + I\sigma^2_e$$

The above mentioned animal models were analyzed using BLUPF90 family of programs [29]. First of all, renumbering program RENUMF90 was used for data quality checking and obtaining basic statistics. This program processed the input and created new data, pedigree and parameter files for further analysis. Subsequently, GIBBSF90+ program was executed to obtain marginal posterior distribution for each parameter in the model. Additional interactive inputs included: the chain was run for 2,00,000 iterations with 2000 as burn-in and 50 as thinning interval. This effectively means that a Gibbs chain was run 2,00,000 times with the first 2000 samples discarded and thereafter, one in every 50[th] sample was stored. Further, POST-GIBBSF90 program was run to analyze the stored samples and obtain posterior means, posterior standard deviations and highest posterior density interval (HPDI) for the genetic parameters for each trait. Convergence of the chain generated by Gibbs sampler was diagnosed based on summary parameters like autocorrelation between lags, effective sample size and tests like Geweke diagnostic test [30]. This was further verified by plotting trace plots for the parameters in order to rule out the existence of any trend in the estimates.

Since three different models were analyzed for each trait, the model with the lowest DIC (Deviance Information Criterion) was selected as the best one appropriately partitioning the existing variance of the concerned trait [31]. Posterior estimates of (co)variance, heritability and other genetic parameters were deduced based on the best fitted model.

## Results and discussion

### Partitioning of variance and posterior heritability estimates

The posterior estimates of (co)variance and heritability for the traits using different models are presented in Tables 2 and 3. Based on the findings, it can be assumed that maternal genetic and permanent environmental effects of dam contribute an important source of variation in the population [32, 33]. Maternal contribution in the expression of birth weight and early growth traits in the progeny has already been widely documented [16, 18, 34]. It is regulated partly by the additive genetic component of the dam and partly by the permanent environment provided by the mother to her offspring [35]. The effect of maternal inheritance on egg sizes produced by the progeny has also been reported [36]. Maternal physiology and permanent

**Table 2. Posterior estimates of variance components and heritability for ASM and body weight traits using different models.**

| Parameters | $\sigma^2_a$ | $\sigma^2_{mat}$ | $\sigma_{am}$ | $\sigma^2_{mpe}$ | $\sigma^2_e$ | $h^2_{additive} \pm$ S.D. | HPDI$_{additive}$ | $h^2_{mat} \pm$ S.D. | $c^2_{mpe} \pm$ S.D. | DIC |
|---|---|---|---|---|---|---|---|---|---|---|
| **ASM** | | | | | | | | | | |
| Model 1 | 15.64 | - | - | - | 63.76 | 0.196±0.049 | 0.104–0.296 | - | - | 7822.304 |
| Model 2 | 26.51 | 7.16 | -9.28 | - | 60.89 | 0.31±0.13 | 0.09–0.58 | 0.083±0.043 | - | 6900.79 |
| **Model 3** | **27.26** | **5.87** | **-9.52** | **1.71** | **60.11** | **0.32±0.13** | **0.07–0.55** | **0.07±0.04** | **0.02±0.02** | **6898.27** |
| **BW0** | | | | | | | | | | |
| **Model 1** | **8.59** | **-** | **-** | **-** | **3.17** | **0.73±0.09** | **0.55–0.89** | **-** | **-** | **4273.69** |
| Model 2 | 2.02 | 5.93 | -2.08 | - | 5.83 | 0.17±0.14 | 0.008–0.44 | 0.51±0.14 | - | 4628.29 |
| Model 3 | 2.58 | 3.44 | -1.92 | 2.15 | 5.47 | 0.22±0.14 | 0.01–0.50 | 0.29±0.16 | 0.18±0.10 | 4597.47 |
| **BW16** | | | | | | | | | | |
| Model 1 | 3765.8 | - | - | - | 7797.8 | 0.32±0.005 | 0.22–0.44 | - | - | 12824.21 |
| **Model 2** | **4091.7** | **612.25** | **-589.68** | **-** | **7570** | **0.35±0.10** | **0.16–0.55** | **0.05±0.04** | **-** | **12813.1** |
| Model 3 | 4020.4 | 609.25 | -625 | 172.31 | 7545.5 | 0.34±0.11 | 0.14–0.54 | 0.052±0.03 | 0.02±0.01 | 12814.82 |
| **BW20** | | | | | | | | | | |
| Model 1 | 5567.5 | - | - | - | 9020.9 | 0.38±0.07 | 0.26–0.51 | - | - | 12544.52 |
| Model 2 | 6139.5 | 1354.7 | -1190.5 | - | 8484 | 0.41±0.13 | 0.18–0.67 | 0.09±0.05 | - | 12515.3 |
| **Model 3** | **6569.5** | **1060.6** | **-1420.4** | **389.69** | **8250.5** | **0.44±0.15** | **0.16–0.74** | **0.07±0.04** | **0.03±0.02** | **12499.51** |
| **BW40** | | | | | | | | | | |
| Model 1 | 16654 | - | - | - | 20011 | 0.45±0.06 | 0.33–0.58 | - | - | 13024.48 |
| Model 2 | 20680 | 4476.4 | -4406 | - | 16892 | 0.55±0.15 | 0.24–0.84 | 0.12±0.05 | - | 12915.04 |
| **Model 3** | **21223** | **3397.1** | **-4695** | **1486.4** | **16379** | **0.56±0.16** | **0.25–0.86** | **0.09±0.05** | **0.04±0.03** | **12891.84** |
| **BW52** | | | | | | | | | | |
| Model 1 | 14937 | - | - | - | 20496 | 0.42±0.07 | 0.28–0.56 | - | - | 9279.58 |
| Model 2 | 13045 | 2584.7 | -299.32 | - | 20881 | 0.36±0.14 | 0.13–0.66 | 0.07±0.05 | - | 9275.76 |
| **Model 3** | **14235** | **2155.7** | **-1086** | **1056.5** | **20185** | **0.39±0.16** | **0.11–0.71** | **0.06±0.04** | **0.03±0.02** | **9263.07** |

($\sigma^2_a$: Additive genetic variance; $\sigma^2_{mat}$: maternal genetic variance; $\sigma_{am}$: covariance between additive genetic and maternal genetic effects; $\sigma^2_{mpe}$: maternal permanent environmental variance; $h^2_{additive} \pm$ S.D.: Posterior direct additive heritability mean ± Posterior standard deviation; HPDI$_{additive}$: Highest Posterior Density Interval for heritability; $h^2_{mat} \pm$ S.D.: Posterior maternal heritability mean ± Posterior standard deviation; $c^2_{mpe} \pm$ S.D.: Posterior maternal permanent environmental effect mean ± Posterior standard deviation; DIC: Deviance Information Criterion)

environment can modulate offspring development and performance through egg traits like egg weight and composition [14].

**Age at sexual maturity.** ASM is an important trait regulating egg production and fertility in chickens [37, 38]. Also, since body weight crucially regulates the attainment of sexual maturity, ASM can be touted as a trait with important implications on almost all the major economic traits in layers [39]. Based on our findings, ASM is medium heritable with the direct additive $h^2$ for the trait in the best model (Model 3) being 0.32±0.13. This means that the trait variation is partly regulated by genes and partly by environment [40]. Interestingly, in model 1, where only additive genetic variance was considered as a source of variation, $h^2$ for the trait was comparatively very low (0.196±0.049). Inclusion of maternal sources of variation substantially improved the goodness of fit of the model and $h^2$ estimate, thus highlighting the importance of maternal inheritance in trait regulation [16].

**Body weight traits.** Body weight traits exert significant influence on the egg production performance and egg quality traits during the laying cycle [41, 42]. Our findings revealed that models incorporating either maternal genetic effect (Model 2) or maternal genetic and permanent environmental effects (Model 3) were the best fitted models for all the traits except BW0 or birth weight. Model 1 was adjudged as the best model for BW0 and the direct additive

**Table 3. Posterior estimates of variance components and heritability for egg weight and egg production traits using different models.**

| Parameters | $\sigma^2_a$ | $\sigma^2_{mat}$ | $\sigma_{am}$ | $\sigma^2_{mpe}$ | $\sigma^2_e$ | $h^2_{additive}$ ± S.D. | HPDI$_{additive}$ | $h^2_{mat}$ ± S.D. | $c^2_{mpe}$ ± S.D. | DIC |
|---|---|---|---|---|---|---|---|---|---|---|
| **EW28** | | | | | | | | | | |
| Model 1 | 2.27 | - | - | - | 6.92 | 0.25±0.06 | 0.13–0.36 | - | - | 5170.42 |
| Model 2 | 3.08 | 1.51 | -1.54 | - | 6.23 | 0.33±0.12 | 0.13–0.58 | 0.16±0.05 | - | 5132.39 |
| Model 3 | 3.09 | 1.29 | -1.46 | 0.23 | 6.19 | 0.33±0.12 | 0.12–0.58 | 0.14±0.06 | 0.02±0.02 | 5129.64 |
| **EW40** | | | | | | | | | | |
| Model 1 | 3.44 | - | - | - | 7.43 | 0.32±0.06 | 0.19–0.44 | - | - | 4768.32 |
| Model 2 | 4.29 | 0.92 | -1.01 | - | 6.84 | 0.39±0.13 | 0.13–0.63 | 0.08±0.05 | - | 4738.58 |
| Model 3 | 4.29 | 0.86 | -1.06 | 0.33 | 6.71 | 0.38±0.13 | 0.12–0.64 | 0.08±0.04 | 0.03±0.02 | 4732.19 |
| **EW52** | | | | | | | | | | |
| Model 1 | 8.77 | - | - | - | 7.63 | 0.53±0.09 | 0.36–0.70 | - | - | 3489.99 |
| Model 2 | 12.29 | 1.23 | -2.35 | - | 5.81 | 0.71±0.21 | 0.32–1.00 | 0.07±0.05 | - | 3289.36 |
| Model 3 | 12.66 | 1.22 | -2.59 | 0.32 | 5.55 | 0.73±0.22 | 0.29–1.00 | 0.07±0.04 | 0.02±0.02 | 3238.35 |
| **EP24** | | | | | | | | | | |
| Model 1 | 10.49 | - | - | - | 41.2 | 0.20±0.05 | 0.11–0.31 | - | - | 6398.87 |
| Model 2 | 15.09 | 2.86 | -4.11 | - | 38.64 | 0.29±0.11 | 0.08–0.50 | 0.05±0.03 | - | 6385.25 |
| Model 3 | 13.99 | 2.49 | -3.79 | 0.96 | 38.89 | 0.26±0.12 | 0.35–0.47 | 0.05±0.03 | 0.02±0.02 | 6389.27 |
| **EP28** | | | | | | | | | | |
| Model 1 | 20.35 | - | - | - | 109.98 | 0.18±0.05 | 0.09–0.29 | - | - | 7826.49 |
| Model 2 | 21.35 | 6.81 | -4.82 | - | 87.85 | 0.19±0.10 | 0.02–0.37 | 0.06±0.04 | - | 7825.17 |
| Model 3 | 19.82 | 5.96 | -5.16 | 3.99 | 86.96 | 0.18±0.10 | 0.05–0.37 | 0.05±0.03 | 0.36±0.25 | 7823.42 |
| **EP40** | | | | | | | | | | |
| Model 1 | 69.83 | - | - | - | 268.15 | 0.21±0.06 | 0.10–0.32 | - | - | 8774.69 |
| Model 2 | 67.63 | 26.44 | -15.59 | - | 264.1 | 0.19±0.10 | 0.01–0.39 | 0.08±0.04 | - | 8773.63 |
| Model 3 | 62.82 | 20.22 | -14.36 | 11.51 | 263.51 | 0.18±0.09 | 0.008–0.36 | 0.06±0.04 | 0.03±0.03 | 8773.73 |
| **EP52** | | | | | | | | | | |
| Model 1 | 60.63 | - | - | - | 340.66 | 0.15±0.07 | 0.04–0.28 | - | - | 5884.52 |
| Model 2 | 79.96 | 35.07 | -34.28 | - | 328.77 | 0.19±0.13 | 0.001–0.43 | 0.08±0.05 | - | 5881.54 |
| Model 3 | 71.72 | 32.88 | -33.96 | 14.81 | 327.47 | 0.17±0.12 | 0.0004–0.41 | 0.08±0.05 | 0.03±0.03 | 5882.44 |
| **EP64** | | | | | | | | | | |
| Model 1 | 59.5 | - | - | - | 597.83 | 0.09±0.06 | 0.003–0.19 | - | - | 5615.17 |
| Model 2 | 33 | 66.61 | -20.37 | - | 589.17 | 0.05±0.05 | 0.0002–0.15 | 0.09±0.05 | - | 5614.91 |
| Model 3 | 36.8 | 61.41 | -25.65 | 20.74 | 580.61 | 0.05±0.05 | 0.00001–0.16 | 0.09±0.05 | 0.03±0.03 | 5616.3 |
| **EP72** | | | | | | | | | | |
| Model 1 | 85.78 | - | - | - | 736.86 | 0.10±0.06 | 0.0001–0.22 | - | - | 5203.88 |
| Model 2 | 62.35 | 117.6 | -48.48 | - | 704.76 | 0.07±0.07 | 0.0006–0.21 | 0.14±0.07 | - | 5198.97 |
| Model 3 | 79.29 | 109.83 | -67.84 | 32.98 | 689.32 | 0.09±0.07 | 0.0002–0.23 | 0.13±0.07 | 0.04±0.03 | 5198.05 |

($\sigma^2_a$: Additive genetic variance; $\sigma^2_{mat}$: maternal genetic variance; $\sigma_{am}$: covariance between additive genetic and maternal genetic effects; $\sigma^2_{mpe}$: maternal permanent environmental variance; $h^2_{additive}$ ± S.D.: Posterior direct additive heritability mean ± Posterior standard deviation; HPDI$_{additive}$: Highest Posterior Density Interval for heritability; $h^2_{mat}$ ± S.D.: Posterior maternal heritability mean ± Posterior standard deviation; $c^2_{mpe}$ ± S.D.: Posterior maternal permanent environmental effect mean ± Posterior standard deviation; DIC: Deviance Information Criterion)

heritability obtained was quite high (0.73±0.09) whereas in the other two models, it was low (Model 2: 0.17±0.14) to medium (Model 3: 0.22±0.14) heritable. HPDI depicting the 95% credible interval in a posterior distribution was also high (0.55–0.89) for the trait. This is quite contrary to the broader understanding that chick birth weight is highly influenced by maternal effects and there is a significant maternal component of variation regulating the trait [43].

However, our findings may be considered in the light of model selection criterion (i.e. DIC) in that though DIC effectively balances model accuracy against complexity (determined by effective number of parameters in a model) by penalizing the loss function, some workers have reported better methods for Bayesian model comparisons [44–46].

Rest of the body weight traits also exhibited medium to high posterior heritability means with BW20 (0.44±0.15) and BW40 (0.56±0.16) reporting higher values. This is in agreement with the fact that growth traits are generally medium to highly heritable and respond very well to genetic selection. Chandan and co-workers [40] reported exactly similar $h^2$ estimate for BW20 (0.443±0.143) in the same layer line based on two generations data (2015–17) using REML approach. Almost similar $h^2$ estimates (0.45±0.13) for the trait were also reported by Manjeet and co-workers [47] in a synthetic white leghorn population from India. The estimated $h^2$ for the trait BW40 in our study is higher than those reported by a study for the trait in the same population [40]. This indicates that Bayesian analysis can improve the estimates by reducing the mean square error (due to prior incorporation), hence, it gives higher $h^2$ values when compared to REML analysis [48, 49]. Nonetheless, most of the workers have also reported moderate to high heritability for BW40 in Indian White Leghorn synthetic populations [50, 51]. The value of $h^2$ for BW16 (0.35±0.10) in our study was similar to that obtained by Rosa and coworkers [52] in a white leghorn population. In our study, the trait BW52 was also medium heritable (0.39±0.16) which is in line with the observations made by most of the earlier workers [53, 54].

Overall, there is enough additive genetic variation available in the body weight traits in IWH population which can be exploited (depending on the trait correlations) to achieve the desired effect in the primary trait of selection [55].

**Egg weight traits.** Egg weight plays an important role in deciding the economics of a laying hen and is known to be heavily influenced by egg quality traits like shell thickness, albumen height, albumen weight, yolk index etc. [56, 57]. Based on our findings, the variance of all the egg weight traits can be partitioned into additive genetic, maternal genetic and permanent environmental effects as Model 3 was adjudged as the best fitted model for all these traits. This highlights that maternal genetics and environment mediates offspring fitness through its effect on egg size and weight. Hence, maternal variance can be successfully exploited to improve egg weight in chickens [58, 59].

EW28, in particular, showed maternal heritability of 0.14±0.06, which underlies the importance of maternal inheritance in regulating the trait variance. The trait was medium heritable with direct additive $h^2$ estimate as 0.33±0.12 which was in agreement with the earlier reports in the same population [59]. This indicates the effectiveness of our breeding program since adequate genetic variance for the trait has been maintained over the past several decades.

As far as the trait EW40 was concerned, $h^2$ was estimated as 0.38±0.13 in our population. Previously, researchers have reported much higher estimates for the trait in white leghorn population with Chatterjee and co-workers [60] obtaining an estimate of 0.54±0.19 while Chandan et al. [40] estimated it to be 0.58±0.11. The probable reason for this could be that both the studies did not partition the maternal components of trait variation. On the contrary, Veeramani et al. [61] reported a lower estimate of sire based $h^2$ for the trait (0.27±0.08) in IWN strain of White Leghorn population in India. This could be attributed to the fact that sire model assumes genetic uniformity of the mates and hence, does not takes into account the additive genetic relationships between all the individuals in a population [62].

For EW52, we obtained a very high posterior $h^2$ estimate (0.73±0.22) in IWH line. This indicates that sufficient genetic variability exists for the trait and it may be included as an additional criterion (apart from egg production) in multi-trait selection for better economic returns. Earlier reports on another White Leghorn line IWK reared in the same environment

have obtained much lower estimates (0.18±0.08) [51]. The probable reason could be that IWK line has been selected for egg weight along with egg numbers due to which additive variance for the trait may have been exhausted due to selection [63].

**Egg production traits.** Most of the egg production traits (except EP24) reported lower heritability estimates. This was quite expected given the fact that egg production traits are generally considered as low to medium heritable. Early production traits like EP24 showed medium heritability estimates (0.29±0.11) which was in line with the findings that heritability in the initial phase of production cycle was relatively high [64, 65]. Traits EP40 and EP52 signifying almost the middle phase of the laying cycle had similar heritabilities (0.19±0.10 and 0.19±0.13 respectively). It further hints at the possibility of analyzing egg production by dividing it into three phases *viz.*, early, middle and later phases of laying cycle due to high genetic correlations and similar heritabilities between the consecutive traits [66].

EP64 and EP72 traits reported very low additive heritability with higher posterior standard deviations (0.05±0.05 and 0.09±0.07 respectively). Since selection is being carried out for later egg production (EP64) in the population, it may have contributed to the reduced $h^2$ estimates. Also, only two generations data was available with respect to the later traits like EP64 and EP72 which may have reduced the precision of the estimates. Therefore, these findings can be further improved by incorporating data for more generations.

Reported literature suggests that direct additive heritability is higher during the initial and later stages of production cycle and comparatively lesser during the peak production period or middle phase of laying cycle [64]. On contrary to that, we found that there is almost a linearly declining trend of heritability across the laying cycle. The probable reason could be that studies reporting higher $h^2$ estimates in the later period of the laying cycle have used random regression models for fitting the egg production data. Random regression models are better suited for fitting longitudinal traits like egg production because they allow studying the trait variation as a function of time [67].

Models incorporating maternal effects (along with direct additive effects) were found to be the most accurate in partitioning trait variance for egg production traits. Model 2 was identified as the best model for all the egg production traits except EP28 and EP72 for which Model 3 was the best. For EP28, the role of maternal permanent environment in influencing egg production was very high (0.36±0.25). Traits EP40 and EP52 showed similar maternal $h^2$ estimates as 0.08±0.04 and 0.08±0.05 respectively. Similarly, for EP64 and EP72 traits, maternal $h^2$ (0.09±0.05 and 0.13±0.07 respectively) was higher than the additive heritability estimates. This may appear on account of adaptive maternal effects which tend to improve the fitness of the offspring. Maternal environment can play a major role in the phenotype of offspring fitness [68, 69].

The complex interplay of direct additive, maternal genetic and maternal environmental variation in different economic traits of poultry can result in the final phenotypic expression of a trait and hence, all of these must be included in the prediction models [70].

## Posterior genetic and environmental correlations

The posterior means of additive genetic and environmental correlations between all the traits have been presented in Tables 4–6. Additive genetic correlation was positive between all the egg production traits with high and strong positive correlation existing between consecutive traits. For instance, EP24 exhibited high positive additive genetic correlation with EP28 (0.89±0.05) which reduced to almost nil between EP24 and EP72 (0.06±0.25). Traits EP64 and EP72 showed strong and highly positive genetic correlation (0.96±0.02) which indicates that former is a perfect indicator of the overall laying ability of the bird in a cycle. Although EP52

**Table 4. Posterior means of additive genetic and environmental correlation between egg production and egg weight traits.**

|  | EP24 | EP28 | EP40 | EP52 | EP64 | EP72 | EW28 | EW40 | EW52 |
|---|---|---|---|---|---|---|---|---|---|
| EP24 |  | 0.89±0.05 | 0.50±0.19 | 0.37±0.21 | 0.03±0.26 | 0.06±0.25 | -0.36±0.17 | -0.34±0.18 | -0.51±0.15 |
| EP28 | 0.84±0.01 |  | 0.73±0.12 | 0.43±0.19 | 0.25±0.24 | 0.24±0.24 | -0.38±0.19 | -0.33±0.19 | -0.31±0.19 |
| EP40 | 0.39±0.05 | 0.56±0.03 |  | 0.87±0.07 | 0.76±0.11 | 0.71±0.13 | -0.34±0.18 | -0.38±0.17 | -0.59±0.14 |
| EP52 | 0.32±0.05 | 0.47±0.04 | 0.88±0.01 |  | 0.89±0.06 | 0.79±0.11 | -0.54±0.29 | -0.84±0.16 | -0.74±0.22 |
| EP64 | 0.26±0.06 | 0.37±0.05 | 0.76±0.03 | 0.92±0.01 |  | 0.96±0.02 | -0.64±0.31 | -0.79±0.23 | -0.67±0.34 |
| EP72 | 0.21±0.06 | 0.31±0.05 | 0.70±0.03 | 0.86±0.02 | 0.97±0.004 |  | -0.60±0.33 | -0.81±0.19 | -0.76±0.25 |
| EW28 | -0.15±0.06 | -0.14±0.06 | -0.09±0.05 | -0.06±0.06 | -0.02±0.07 | -0.06±0.07 |  | 0.91±0.08 | 0.91±0.07 |
| EW40 | -0.46±0.07 | -0.06±0.07 | -0.09±0.06 | -0.05±0.07 | -0.08±0.07 | -0.05±0.07 | 0.33±0.06 |  | 0.96±0.03 |
| EW52 | 0.04±0.09 | -0.09±0.09 | 0.06±0.09 | -0.13±0.09 | -0.15±0.10 | -0.15±0.09 | 0.08±0.09 | 0.27±0.09 |  |

Values above the diagonal indicate posterior additive genetic correlation between the traits whereas values below the diagonal signify posterior environmental correlation between the traits

**Table 5. Posterior means of additive genetic and environmental correlation between egg production and body weight traits.**

|  | ASM | BW1 | BW16 | BW20 | BW40 | BW52 | EP24 | EP28 | EP40 | EP52 | EP64 | EP72 |
|---|---|---|---|---|---|---|---|---|---|---|---|---|
| ASM |  | 0.32±0.10 | -0.12±0.14 | 0.03±0.14 | 0.26±0.14 | 0.32±0.13 | -0.59±0.19 | -0.84±0.07 | 0.08±0.29 | -0.41±0.20 | 0.23±0.28 | 0.24±0.28 |
| BW1 | -0.54±0.13 |  | 0.32±0.09 | 0.47±0.10 | 0.37±0.08 | 0.32±0.09 | -0.54±0.19 | 0.10±0.21 | -0.89±0.10 | -0.51±0.16 | -0.93±0.09 | -0.93±0.08 |
| BW16 | -0.42±0.04 | -0.24±0.11 |  | 0.89±0.05 | 0.68±0.09 | 0.75±0.07 | -0.31±0.16 | -0.33±0.15 | -0.15±0.18 | -0.24±0.19 | 0.08±0.19 | 0.12±0.19 |
| BW20 | -0.23±0.05 | -0.59±0.12 | 0.52±0.04 |  | 0.87±0.05 | 0.90±0.03 | -0.20±0.17 | -0.15±0.17 | -0.19±0.18 | -0.06±0.21 | 0.11±0.18 | 0.14±0.19 |
| BW40 | -0.09±0.06 | -0.44±0.09 | 0.28±0.06 | 0.44±0.05 |  | 0.96±0.01 | -0.28±0.16 | -0.35±0.16 | -0.15±0.17 | -0.32±0.19 | 0.006±0.18 | 0.01±0.18 |
| BW52 | -0.14±0.06 | -0.37±0.11 | 0.25±0.05 | 0.38±0.05 | 0.65±0.03 |  | -0.37±0.16 | -0.35±0.16 | -0.29±0.16 | -0.30±0.19 | -0.04±0.18 | -0.009±0.18 |
| EP24 | -0.62±0.03 | 0.27±0.05 | 0.45±0.06 | 0.18±0.06 | 0.02±0.07 | 0.04±0.07 |  | 0.89±0.05 | 0.50±0.19 | 0.37±0.21 | 0.03±0.26 | 0.06±0.25 |
| EP28 | -0.49±0.03 | -0.12±0.18 | 0.39±0.05 | 0.29±0.05 | 0.12±0.06 | 0.13±0.07 | 0.84±0.01 |  | 0.73±0.12 | 0.43±0.19 | 0.25±0.24 | 0.24±0.24 |
| EP40 | -0.19±0.05 | 0.47±0.05 | 0.18±0.06 | 0.15±0.06 | 0.14±0.07 | -0.01±0.08 | 0.39±0.05 | 0.56±0.03 |  | 0.87±0.07 | 0.76±0.11 | 0.71±0.13 |
| EP52 | -0.28±0.04 | 0.35±0.15 | 0.18±0.06 | 0.15±0.06 | 0.11±0.07 | 0.06±0.08 | 0.32±0.05 | 0.47±0.04 | 0.88±0.01 |  | 0.89±0.06 | 0.79±0.11 |
| EP64 | -0.12±0.05 | 0.49±0.05 | 0.07±0.06 | 0.08±0.07 | 0.06±0.07 | -0.09±0.08 | 0.26±0.06 | 0.37±0.05 | 0.76±0.03 | 0.92±0.01 |  | 0.96±0.02 |
| EP72 | -0.09±0.05 | 0.52±0.05 | 0.05±0.06 | 0.08±0.07 | 0.04±0.07 | -0.11±0.08 | 0.21±0.06 | 0.31±0.05 | 0.70±0.03 | 0.86±0.02 | 0.97±0.004 |  |

Values above the diagonal indicate posterior additive genetic correlation between the traits whereas values below the diagonal signify posterior environmental correlation between the traits

**Table 6. Posterior means of additive genetic and environmental correlation between body weight, ASM and egg weight traits.**

|  | ASM | BW0 | BW16 | BW20 | BW40 | BW52 | EW28 | EW40 | EW52 |
|---|---|---|---|---|---|---|---|---|---|
| ASM |  | 0.32±0.10 | -0.12±0.14 | 0.03±0.14 | 0.26±0.14 | 0.32±0.13 | 0.32±0.17 | 0.38±0.15 | 0.17±0.17 |
| BW0 | -0.54±0.13 |  | 0.32±0.09 | 0.47±0.10 | 0.37±0.08 | 0.32±0.09 | 0.79±0.19 | 0.86±0.14 | 0.86±0.14 |
| BW16 | -0.42±0.04 | -0.24±0.11 |  | 0.89±0.05 | 0.68±0.09 | 0.75±0.07 | 0.49±0.13 | 0.54±0.11 | 0.77±0.09 |
| BW20 | -0.23±0.05 | -0.59±0.12 | 0.52±0.04 |  | 0.87±0.05 | 0.90±0.03 | 0.58±0.13 | 0.73±0.15 | 0.65±0.11 |
| BW40 | -0.09±0.06 | -0.44±0.09 | 0.28±0.06 | 0.44±0.05 |  | 0.96±0.01 | 0.59±0.11 | 0.72±0.07 | 0.61±0.09 |
| BW52 | -0.14±0.06 | -0.37±0.11 | 0.25±0.05 | 0.38±0.05 | 0.65±0.03 |  | 0.37±0.14 | 0.62±0.09 | 0.57±0.11 |
| EW28 | 0.06±0.05 | -0.18±0.06 | -0.02±0.06 | 0.08±0.07 | 0.13±0.07 | 0.13±0.07 |  | 0.91±0.08 | 0.91±0.07 |
| EW40 | -0.02±0.06 | -0.49±0.04 | -0.08±0.07 | -0.25±0.09 | -0.009±0.08 | -0.009±0.09 | 0.33±0.06 |  | 0.96±0.03 |
| EW52 | -0.04±0.08 | -0.93±0.009 | -0.25±0.11 | -0.06±0.12 | -0.09±0.12 | -0.01±0.15 | 0.08±0.09 | 0.27±0.09 |  |

Values above the diagonal indicate posterior additive genetic correlation between the traits whereas values below the diagonal signify posterior environmental correlation between the traits

also has high positive correlation with EP72 (0.79±0.11), it is still not large enough to choose it as a selection criteria for overall laying ability of the bird. Chandan et al. [40] advocated selecting for higher egg production based on part period egg production of EP52 due to very strong correlation between EP52 and EP64 (0.926±0.050). However, they did not calculate the genetic correlation between EP52 and EP72. In our study, environmental correlation between egg production traits was medium to high between different traits with EP40 and subsequent traits witnessing high correlation values.

Egg weight traits also witnessed very high and positive genetic correlations (>0.90) between the traits. This essentially means that we can consider selecting for an early egg weight trait (say EW28) for higher returns in the future breeding programs. This is also in line with the findings by previous workers that genetic correlation between egg weights at different ages was very high, especially between consecutive time periods [71, 72]. However, Chandan et al. [40] have obtained comparatively higher estimates between EW28, EW40 and EW52 traits in the same IWH population. This may be due to the reason that our study also included maternal effects in the model which may have partitioned all the genetic parameters amongst different components.

Egg weights were negatively correlated genetically with all the egg production traits with the correlation becoming more strong and negative towards the end of the laying cycle. This is in agreement with the already known trade-off between egg size and egg production in that consistent selection for higher egg numbers diverts the resources or energy available for producing larger sized eggs [73]. Dam or maternal effect is also an important factor regulating the dynamics of egg production and egg size in females [74]. Since both the set of traits are economically important, it is advised to select for these traits with the help of a selection index according appropriate weightages to all the traits [71].

ASM was negatively correlated genetically with early egg production traits like EP24 (-0.59 ±0.19), EP28 (-0.84±0.07) and also with EP52 (-0.41±0.20) while it was having a positive correlation with EP64 (0.23±0.28) and EP72 (0.24±0.28). It is widely known that an antagonistic relationship exists between ASM and egg production traits [9, 75]. The large posterior standard deviation component in the genetic correlation values of ASM with the late EP traits (EP64, EP72) is reflective of the need to increase the population size for getting more accurate genetic picture of relationship between the traits. ASM and BW0 had moderate positive genetic correlation which is obvious given the fact that higher birth weight leads to early sexual maturity due to high positive genetic correlation between the BW0 and body weight at sexual maturity [39, 76]. ASM showed positive genetic correlation with the later body weight traits including BW20, BW40 and BW52. It may be on account of the fact that ASM is a trait related to sexual maturity and shares negative correlation with body weight at sexual maturity. However, a positive correlation may exist between ASM and later body weight traits as the late maturing birds have lower egg production and higher body weights due to the existing negative genetic correlation between body weight and production traits [40]. ASM was positively and relatively moderately correlated with all the egg weight traits due to the negative genetic relationship of these traits with egg production as higher ASM will decrease total egg production and ultimately, egg size will increase [73, 75].

In agreement with previous studies, there was a strong positive relationship amongst the body weight traits in our study [76]. The correlation between BW0 and rest of the body weight traits was comparatively less than other growth combinations due to the reason that chick birth weight has a strong maternal influence (genetic as well as permanent environmental effect of dam) [58]. However, a negative genetic correlation existed between different egg production traits and body weight traits. This is a reiteration of the oft-repeated remark that lighter hens are more efficient egg producers and that intense selection for egg production

introduces severe depression on body weight traits in layers [41, 77]. As far as the body weight and egg weight traits were concerned, there was high positive genetic correlation between all the traits. This may be due to obvious physiological reasons and is in line with the observations made by most of the workers [71, 78].

The future breeding program of IWH population should be designed considering the existing additive variability of traits and the genetic correlations obtained in this study. Also, the maternal components of variation constitute an important source of variation and should be included along with additive variance for accurate estimation of genetic parameters for all the traits. Simultaneously, genetic parameters should be re-estimated after every completed generation in order to observe the underlying changes in trait variation and also to improve the precision of the genetic estimates.

## Conclusions

This study was conducted to delineate the role of maternal genetic and permanent environment in variance partitioning and to estimate the genetic parameters of various economic traits in a population of White Leghorn layer line IWH using Bayesian approach. Based on our findings, we conclude that most of the economic traits are regulated by maternal effects with traits like EW28, EP72 and EP28 having a comparatively larger component of maternal genetic and maternal permanent environmental variance, respectively. Therefore, maternal sources of variation should be included along with additive component for genetic parameter estimation. Although Bayesian inference improved the accuracy of additive genetic variance and direct heritability estimates, smaller population size was still a limitation in getting precise estimates for maternal components of variance. This can be improved upon by increasing the data size of the population and re-evaluation and re-fitting of the models using Bayesian methods.

## Acknowledgments

The authors are thankful to the Director, ICAR-Directorate of Poultry Research, Hyderabad for his constant support and encouragement. The technical services provided by the farm and hatchery staffs are duly acknowledged.

## Author Contributions

**Conceptualization:** Aneet Kour, U. Rajkumar.

**Data curation:** Aneet Kour, K. S. Rajaravindra, L. Leslie Leo Prince, M. Niranjan, B. L. N. Reddy.

**Formal analysis:** Aneet Kour.

**Funding acquisition:** R. N. Chatterjee.

**Investigation:** Aneet Kour, K. S. Rajaravindra, L. Leslie Leo Prince, Santosh Haunshi, M. Niranjan, B. L. N. Reddy, U. Rajkumar.

**Methodology:** Aneet Kour, K. S. Rajaravindra, L. Leslie Leo Prince, Santosh Haunshi, B. L. N. Reddy.

**Project administration:** U. Rajkumar.

**Software:** Aneet Kour, L. Leslie Leo Prince.

**Supervision:** R. N. Chatterjee, U. Rajkumar.

**Validation:** L. Leslie Leo Prince, Santosh Haunshi, M. Niranjan, B. L. N. Reddy, U. Rajkumar.

**Visualization:** M. Niranjan.

**Writing – original draft:** Aneet Kour.

**Writing – review & editing:** U. Rajkumar.

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
