## [Decision Letter · Decision Letter 0]

22 May 2024

PONE-D-24-11008Delineating maternal influence in regulation of variance in major economic traits of White Leghorns: Bayesian insightsPLOS ONE

Dear Dr. Rajkumar,

Thank you for submitting your manuscript to PLOS ONE. After careful consideration, we feel that it has merit but does not fully meet PLOS ONE’s publication criteria as it currently stands. Therefore, we invite you to submit a revised version of the manuscript that addresses the points raised during the review process.

We look forward to receiving your revised manuscript.

Kind regards,

Ewa Tomaszewska, DVM Ph.D

Academic Editor

PLOS ONE

2. In the online submission form, you indicated that your data is available only on request from a third party. Please note that your Data Availability Statement is currently missing the contact details for the third party, such as an email address or a link to where data requests can be made. Please update your statement with the missing information.

Reviewers' comments:

Reviewer's Responses to Questions

**Comments to the Author**

1. Is the manuscript technically sound, and do the data support the conclusions?

Reviewer #1: Yes

2. Has the statistical analysis been performed appropriately and rigorously? 

Reviewer #1: Yes

3. Have the authors made all data underlying the findings in their manuscript fully available?

Reviewer #1: Yes

4. Is the manuscript presented in an intelligible fashion and written in standard English?

Reviewer #1: Yes

5. Review Comments to the Author

Reviewer #1: Dear Authors,

Strictly follow my comments/corrections in attached file, in general:

Introduction: in the end, need of the project should be ideally raised followed by the hypothesis of the study.

Results and Discussion: add logical reasoning of each result before discussing with the previous studies.

Conclusion: add limitations and implications of the study.

Thank You!

6. PLOS authors have the option to publish the peer review history of their article (what does this mean?). If published, this will include your full peer review and any attached files.

Reviewer #1: No

---

## [Author Response · Author response to Decision Letter 0]

12 Jun 2024

S.No. Comment Response

Reviewer’s comments

1. Introduction: in the end, need of the project should be ideally raised followed by the hypothesis of the study.

 As rightly pointed out by the expert, the necessary changes have been made in the lines 88-97 in the file ‘Revised Manuscript with Track Changes’

2. Materials and Methods: which experimental design was followed? 

Full sib design in a pedigreed mating was followed in the study for data generation. The required information has been added in the lines 104-107 in the file ‘Revised Manuscript with Track Changes’

3. Results and Discussion: This section needs improvement. Add logical reasoning of each result before discussing with the previous studies 

 The results and discussion section has been modified as per the suggestion of the reviewer and the changes can be seen in the Results and discussion section in the files ‘Revised Manuscript with Track Changes ‘and Manuscript

4. Conclusions: Add limitations and implications of the study 

As desired by the worthy reviewer, needful has been done in the lines 563-573 in the file ‘Revised Manuscript with Track Changes’

Editor’s comments

 As suggested by the worthy editor, the manuscript has been thoroughly edited to match journal’s requirements

2. Please update your Data availability statement with the missing information. 

As per the journal’s requirements, the data availability statement has been revised and a unique identifier has been provided in the lines 598-608 in the file ‘Revised Manuscript with Track Changes’

3. Please review your reference list to ensure that it is complete and correct………………. 

The references list has been thoroughly screened and all the references have been checked. Accordingly, Reference no. 25 (Lines 691-693) has been modified as a Publisher correction to the previous article has been published. 

Also, some minor correction in the reference style has been made in the Reference no. 41 (Line 741). 

All these changes can be easily seen in the file ‘Revised Manuscript with Track Changes’

---

## [Decision Letter · Decision Letter 1]

16 Jul 2024

Delineating maternal influence in regulation of variance in major economic traits of White Leghorns: Bayesian insights

PONE-D-24-11008R1

Dear Dr. U. Rajkumar,

We’re pleased to inform you that your manuscript has been judged scientifically suitable for publication and will be formally accepted for publication once it meets all outstanding technical requirements.

Kind regards,

Ewa Tomaszewska, DVM Ph.D

Academic Editor

PLOS ONE

Additional Editor Comments (optional):

Reviewers' comments:

Reviewer's Responses to Questions

**Comments to the Author**

1. If the authors have adequately addressed your comments raised in a previous round of review and you feel that this manuscript is now acceptable for publication, you may indicate that here to bypass the “Comments to the Author” section, enter your conflict of interest statement in the “Confidential to Editor” section, and submit your "Accept" recommendation.

Reviewer #1: All comments have been addressed

2. Is the manuscript technically sound, and do the data support the conclusions?

Reviewer #1: Yes

3. Has the statistical analysis been performed appropriately and rigorously? 

Reviewer #1: Yes

4. Have the authors made all data underlying the findings in their manuscript fully available?

Reviewer #1: Yes

5. Is the manuscript presented in an intelligible fashion and written in standard English?

Reviewer #1: Yes

6. Review Comments to the Author

Reviewer #1: (No Response)

7. PLOS authors have the option to publish the peer review history of their article (what does this mean?). If published, this will include your full peer review and any attached files.

Reviewer #1: No

---

## [Editor Report · Acceptance letter]

18 Jul 2024

PONE-D-24-11008R1 

PLOS ONE

Dear Dr. Rajkumar, 

I'm pleased to inform you that your manuscript has been deemed suitable for publication in PLOS ONE. Congratulations! Your manuscript is now being handed over to our production team.

Kind regards, 

on behalf of

Professor Ewa Tomaszewska 

Academic Editor

PLOS ONE